# An Efficient Approach to 2-CF_3_-Indoles Based on *ortho*-Nitrobenzaldehydes

**DOI:** 10.3390/molecules26237365

**Published:** 2021-12-04

**Authors:** Vasiliy M. Muzalevskiy, Zoia A. Sizova, Vladimir T. Abaev, Valentine G. Nenajdenko

**Affiliations:** 1Department of Chemistry, Lomonosov Moscow State University, 119899 Moscow, Russia; muzvas@mail.ru (V.M.M.); syzova@mail.ru (Z.A.S.); 2North Ossetian State University, 44-46 Vatutina St., 362025 Vladikavkaz, Russia; hampazero@mail.ru; 3North Caucasus Federal University, 1a Pushkin St., 355009 Stavropol, Russia

**Keywords:** CF_3_-group, catalytic olefination reaction, nitro group, reduction, indole, fluorine

## Abstract

The catalytic olefination reaction of 2-nitrobenzaldehydes with CF_3_CCl_3_ afforded stereoselectively trifluoromethylated *ortho*-nitrostyrenes in up to 88% yield. The reaction of these alkenes with pyrrolidine permits preparation of α-CF_3_-β-(2-nitroaryl) enamines. Subsequent one pot reduction of nitro-group by Fe-AcOH-H_2_O system initiated intramolecular cyclization to afford 2-CF_3_-indoles. Target products can be prepared in up to 85% yields. Broad synthetic scope of the reaction was shown as well as some followed up transformations of 2- CF_3_-indole.

## 1. Introduction

Indole has been discovered in 1866 by Bayer [1]. This type of heterocycles became an object of intensive investigations [2,3,4,5,6,7,8] and recognized as a “privileged structure” in drug discovery [9]. Indole motif is an important structural unit of many pharmaceuticals and natural products [10]. Seven derivatives of indole can be found in the list of 200 best selling drugs in 2020. Tagrisso ($4.328 Bn), Trikafta ($3.864 Bn), Ofev ($2.448 Bn), Leuprorelin ($1.834 Bn), Alecensa ($1.292 Bn), Zoladex ($ 0.888 Bn) and Sutent ($0.819 Bn) were sold for more than $ 15 billion totally in 2020 worldwide [11].

Chemistry of fluorinated organic compounds is a booming area of modern organic chemistry, which is a result of unique physicochemical as well as biological properties of these compounds [12,13,14,15,16,17,18,19,20,21,22,23]. Thus, about 20% (more than 300 compounds) of currently used drugs [24,25,26,27,28,29,30,31] contain at least one fluorine atom [32]. In 2020, approximately 25% (14 out of 53) of all drugs and about 35% (14 out of 40) of “small-molecule drugs” approved by the FDA are fluorinated compounds [33]. At the same time, about 59% of all small-molecule drugs have a nitrogen heterocyclic motif [10]. Last year revealed, that three out of every four small-molecule drugs approved by the FDA in 2020 (28 out of 37) are representatives of that class [33]. Hence, elaboration of novel pathways to fluorinated nitrogen heterocycles are of great demand [34,35,36,37,38,39,40,41].

The brilliant example of such interest are 2-CF_3_-indoles. According to the Reaxys database, this class of compounds enjoyed a boom of attention last decade. Thus, 107 out of 152 research articles dealing with 2-CF_3_-indoles were published from 2011 to 2021. Previous decade revealed 19 articles, and 26 articles were published in period from 1977 to 2001 [42]. These massive investigations gained several promising bioactive 2-CF_3_-indoles (Figure 1). The 2′-trifluoromethyl analogue of Indomethacin **I** was appeared to be a potent and selective COX-2 inhibitor [43]. Compound **II** having 2-CF_3_-indole and cinnamic amide moieties possess anti-inflammatory and neuroprotective actions [44]. Indolyl-pyridinyl-propenone **III** was found to possess properties of antiproliferative [45] and antineoplastic agent [46]. 2-CF_3_-indole **IV** revealed antifungal properties (Figure 1) [47].

Most synthetic approaches to such indoles can be divided to the methods of direct trifluoromethylation of indole core and cyclizations of various precursors having CF_3_ group in appropriate position [38]. Both radical and electrophilic trifluoromethylations was performed using bis(perfluoroalkanoyl) peroxides [48], difluorodiiodomethane [49], hypervalent iodine reagents [50,51,52,53], CF_3_I [54,55,56,57,58], Umemoto’s reagents [59,60], [(phen)CuCF_3_] [61], and CF_3_SO_2_Na [62,63,64,65]. Cyclization approaches are based on formation of C2-C3 bond as a key step and deal with transformations of compounds having *ortho*-toluidine fragment [66,67,68,69,70,71,72,73,74,75,76]. One work reported transformation of 4- and 6-nitro-1-hydroxy indoles to NH indoles under treatment with bromoacetophenone (Figure 2) [77].

Several years ago, we have elaborated convenient approach to α-CF_3_-β-aryl enamines on the base of the reaction of β-chloro-β-trifluromethylstyrenes with amines [78,79,80]. This potent CF_3_-building blocks were successfully used as synthetic equivalents of trifluoromethyl benzyl ketones in the Fisher and Pictet–Spengler reaction to give 2-CF_3_-3-arylindoles and CF_3_-β-carbolines [81], synthesis of CF_3_-enones [82,83,84] and α-CF_3_-phenethylamines [85]. In continuation of the investigation of synthetic potential of α-CF_3_-β-aryl enamines, we report in this article one pot two step synthesis of 2-CF_3_-indoles (Figure 2).

## 2. Results

First, we investigated olefination of 2-nitrobenzaldehydes **1** to prepare the corresponding trifluoromethylated styrenes **2**. The catalytic olefination reaction (COR) [15,86,87,88,89,90] and Wittig reaction were used for the synthesis of these alkenes. We performed screening of the reaction conditions for COR (see Appendix A). It was found, that ethylene glycol [89] is the solvent of choice for these substrates, in contrast to EtOH traditionally used for COR with CF_3_CCl_3_ [90]. It was also found, that the yield is very sensitive to the nature of the substituents (additional to *ortho*-nitro group) in aryl ring. The best yield in the whole series was obtained for unsubstituted 2-nitrobenzaldehyde, which was transformed by COR to styrene **2a** in 88% yield. In the case of additional alkyl-, alkoxy- and halogen substituents in the aryl ring corresponding styrenes were isolated in good to high yields. However, in the case of aldehydes **1j,l,m** having strong EWG substituents (nitro-, cyano- and carboxymethyl- groups) in 4-position the corresponding alkenes **2j,l,m** were synthesized in lower yields using COR. Therefore, we tried also alternative synthesis based on Wittig olefination. As a result, some improvement was observed for these problematic aldehydes. It should be noted that olefination of 2-nitrobenzaldehydes using both methods proceeds stereoselectively to form mostly *Z*-isomer in up to 96:4 ratio with minor *E*-isomer. Assignment of the configuration of the isomers was maintained by comparison with the literature NMR data of similar styrenes without *ortho*-nitro-group [90] (Figure 1).

Having in hand a series of trifluoromethylated *ortho*-nitrostyrenes, we investigated their transformation to 2-CF_3_-indoles. The treatment of styrenes **2** with an access of pyrrolidine at room temperature led to α-CF_3_-enamines **3** in high yield. We assumed, that reduction of ortho-nitro aryl derived α-CF_3_-enamines **3** could led to 2-CF_3_-indoles **4** through formation of intermediate anilines **3′ [91,92]**. The reduction of model enamine **3a** was studied in various conditions. It was found, that HCO_2_H-Pd/C, Fe-AcOH-H_2_O and Zn-AcOH-H_2_O systems worked well to give 2-CF_3_-indole **3a** in 85, 86 and 85% yield correspondingly according to ^19^F NMR. Although all these systems showed almost equal results, we used Fe-AcOH-H_2_O for our further transformations due to the lower price and toxicity of iron [93]. It should be noted, that crude enamine **3a** can be used directly after evaporation of excessive pyrrolidine. So, the transformation of styrene **2** into indole **4** can be maintained as a one pot reaction without isolation of intermediate enamine **3**. Moreover, this one pot conditions work for multigram scale reaction to afford 3.257 g (72%) of indole **4a** in one run (Figure 2).

Using these optimal conditions, we performed the synthesis of various 2-CF_3_-indoles **4**. It was found that the reaction has a general character allowing to prepare 2-CF_3_-indoles having both electron-donating and electron-withdrawing groups in various positions of indole ring in good to high yield. 6-Amino-2-CF_3_-indole **4j** was synthesized in the case of styrene **2j** having additional nitro-group. This indole is a perspective object for further modifications at amino group, which can provide compounds interesting for the medicinal chemistry.

One can notice that indoles **4** were mostly prepared in the yields higher than 50%, which is high enough taking into account the three step transformation. In contrast, indoles **4e** and **4i** were obtained in moderate yield (43% and 25%). The explanation of that fact is a side process taking place at the step of formation of enamine **3**. Thus, monitoring of the reaction mixture in the reaction of **2c** with pyrrolidine revealed the presence of compound **5c**, which was isolated in 15% yield together with enamine **3c** (80%). The structure of **5c** was assigned by means of NMR and HRMS data. Thus, the key signals of **5c** are the signals of carbonyl group (192.4 ppm), quaternary aminal carbon adjacent to CF_3_-group (quadruplet at 86.4 ppm, *J*_CF_ = 28.1 Hz) in ^13^C NMR and N-OH group (7.74 ppm) in ^1^H NMR. We have also observed formation of similar N-hydroxy indolin-3-ones **5** in several other reactions. Thus, in case of enamines **3e** and **3i** the admixture of compounds **5e** and **5i** were 28% and 39%, correspondingly (by ^19^F NMR; see Appendix A for details). Even in the case of enamine **3a** we observed formation of **5a** in 4% yield (by ^19^F NMR). We did not investigate this side reaction thoroughly, but possible mechanism of this transformation was proposed using the literature data (Figure 3) [94,95]. At first step dehydrochlorination of **2c** leads to alkyne **6** [78]. Next, it is attacked by pyrrolidine to give zwitterion **7**. Proton transfer in **7** affords enamine **3c**. Alternatively, transformation of **7** leads to transfer of oxygen to form nitroso compound **8.** This intermediate has in the structure a strong electron-donating fragment of “enoloenamine”. Intramolecular attack of this fragment to nitroso group led to indolin-1-olate derivative **9.** Its protonation leads to N-hydroxy indolin-3-one **5c.**

Interesting results were obtained in the case of styrenes **2n,o**. These alkenes have halogens in *para*-position to nitro-group, which activates nucleophilic substitution of them. It was found that treatment of 4-fluorostyrene **2o** with pyrrolidine led to substitution of both fluorine and chlorine during 1–2 h to give enamine **3n** in 90% yield (Figure 4). Similarly, substitution of both chlorine atoms in 4-chlorostyrene **2n** afforded enamine **3n** in 72% yield. However, in this case about 2–3 days were needed for full substitution of chlorine adjacent to aryl ring. It is not surprising, because fluorine is a better leaving group than chlorine. Next, we performed one pot synthesis of indole **10a** from 4-fluorostyrene **2o**. As a result, indole **10a** was isolated in 45% yield (Figure 4).

We proposed that using less nucleophilic amines would allow to perform selective synthesis of enamine without substitution of halogen in aryl ring. However, the reaction of 4-fluorostyrene **2o** with piperidine afforded a mixture of enamine **11a** and styrene **2p** at room temperature. The heating of this reaction mixture at 90 °C for 3 h led to selective transformation of **2p** into **11a** (by ^19^F NMR), which was converted into indole **10b** in 44% yield (one-pot). To our delight, the reaction of 4-chlorostyrene **2n** with piperidine proceeded only at the double bond to form enamine **11b** (observed in ^19^F NMR) after 1h at room temperature. One pot transformation of **11b** under standard conditions afforded 5-chloro-2-CF_3_-indole **4n** in total 71% yield (Figure 5).

To investigate the scope of the synthesis of 5-amino substituted indoles, we performed several reactions of styrene **2o** with other primary and secondary amines. As a result, new family of 2-CF_3_-indoles **10c–g** having amine fragments of morpholine, azepane, diethylamine, methylamine and *n*-hexylamine was synthesized in good yields (Figure 6).

Having prepared a set of 2-CF_3_-indoles we found surprisingly that many typical reactions known for indoles are unknown for 2-CF_3_-indoles. To fill this gap, we maintained reactions of indole **4a** with several C-centered electrophiles. In our hands, formylation reaction by POCl_3_-DMF afforded 3-formyl-2-CF_3_-indole **17** in 53% yield. Friedel–Crafts acylation with AcCl-AlCl_3_ led to corresponding ketone **18** in 64% yield. Reaction with ethoxy CF_3_-enone **19** under catalysis with BF_3_·Et_2_O gave α,β-unsaturated CF_3_ ketone **20**, which is a valuable building block for the synthesis of complex fluorinated molecules. Very interesting results were observed in the reactions of 2-CF_3_-indole with arylaldehydes in the media of alcohols under catalysis with MeSO_3_H. The reaction with benzaldehyde, 4-chloro- and 4-methoxybenzaldehydes in methanol afforded methoxy-derivatives **21** in good yields. The reaction with 1.2 equivalents of benzaldehyde in ethanol led to ethoxy-derivative **22** in 74% yield, while the reaction with 0.5 equivalents of benzaldehyde in ethanol resulted in bisindolylmethane derivative **23** in moderate yield (Figure 7). NMR monitoring of the reaction revealed, that after first few hours both indoles **22** and **23** can be found in the reaction mixture. Further heating led to decreasing of the amount of 22, while the amount of **23** showed increase. Based on that fact, we rationalized possible mechanism of formation of **23** as follows. At first step, **4a** reacts with aldehyde to form 22, which is protonated by strong methanesulfonic acid to give oxonium salt **24**. Friedel–Crafts alkylation of indole **4a** by this oxonium salt afforded bisindolylmethane derivative **23**.

It should be noted that a lot of attention has been paid to the elaboration of novel strategies for the synthesis of bisindolylmethane derivatives, because many of them exhibit a various kinds of physiological activity [96,97,98,99]. Thus, bisindolylmethanes revealed properties of antibacterial, antifungal, antimicrobial, anti-inflammatory and anti-cancer agents [100,101,102,103,104,105]. In addition, this structural unit can be found in the natural sources, for example in marine alkoloids [106,107,108]. To the best of our knowledge fluorinated bisindolylmethanes have not been reported to date. We believe that our approach to these compounds can be useful in design of potentially active physiologically active compounds.

## 3. Materials and Methods

**General remarks.**^1^H, ^13^C and ^19^F NMR spectra were recorded on Bruker AVANCE 400 MHz spectrometer in CD_3_CN, DMSO-*d_6_* and CDCl_3_ at 400, 100 and 376 MHz, respectively. Chemical shifts (*δ*) in ppm are reported with the use of the residual CHD_2_CN, DMSO-*d*_5_ and chloroform signals (1.94, 2.54 and 7.25 for ^1^H and 1.30, 39.5 77.0 for ^13^C) as internal reference. The ^19^F chemical shifts were referenced to C_6_F_6_, (−162.9 ppm). The coupling constants (*J*) are given in Hertz (Hz). ESI-MS spectra were measured at MicroTof Bruker Daltonics instrument. TLC analysis was performed on “Merck 60 F_254_” plates. Column chromatography was performed on silica gel. All reagents were of reagent grade and were used as such or were distilled prior to use. β-Chloro-β-trifluoromethylstyrenes **1** were prepared as reported previously by catalytic olefination reaction [89,90] or by Wittig reaction [109]. Melting points were determined on an Electrothermal 9100 apparatus.

**Synthesis of styrenes 2 by catalytic olefination reaction in EtOH or DMSO (general procedure I, 5 mmol scale) [90].** One neck 100 mL round bottomed flask was charged with N_2_H_4_·H_2_O (0.265 g, 5.25 mmol), and solution of corresponding benzaldehyde (5 mmol in 25 mL of EtOH or DMSO) was added and stirred for 3 h until aldehyde disappeared (TLC control). Next, 1,2-ethylenediamine (0.65 mL, 7.5 mmol), CuCl (0.050 g, 0.5 mmol) were added and stirred for 1–2 min. After that CF_3_CCl_3_ (1.78 mL, 15 mmol) was added in one portion at cooling by cold water bath. Reaction mixture stirred overnight at room temperature, poured into water (100 mL) and extracted with CH_2_Cl_2_ (3 × 20 mL). Combined extract was washed with water (20 mL) and dried over Na_2_SO_4_. Solvents were evaporated in vacuo, the residue was purified by passing through a short silica gel pad using 3:1 mixture of hexane and CH_2_Cl_2_ as an eluent.

**Synthesis of styrenes 2 by catalytic olefination reaction in ethylene glycol (general procedure II) [89].** One neck 50 mL round bottomed flask was charged with 1 mmol of corresponding benzaldehyde, 10 mL of ethylene glycol, 0.25 mL (5 mmol) of N_2_H_4_·H_2_O and stirred 0.5–1h until aldehyde disappeared (TLC control). Next, 0.38 mL (4.4 mmol) of 1,2-ethylenediamine, 0.0086 g (0.05 mmol) of CuCl_2_·2H_2_O was added and stirred for 1–2 min. After that CF_3_CCl_3_ (0.71 mL, 6 mmol) was added in one portion at cooling by cold water bath. Reaction mixture stirred overnight at room temperature, poured into water (50 mL) and extracted with CH_2_Cl_2_ (3 × 20 mL). Combined extract was washed with water (20 mL) and dried over Na_2_SO_4_. Solvents were evaporated in vacuo, the residue was purified by passing through a short silica gel pad using 3:1 mixture of hexane and CH_2_Cl_2_ as an eluent.

**Synthesis of styrene 2a by catalytic olefination reaction in EtOH (150 mmol scale).** One neck 1000 mL round bottomed flask was charged with N_2_H_4_·H_2_O (5.25 g, 105 mmol), and solution of 2-nitrobenzaldehyde (15.11 g, 100 mmol in 175 mL of EtOH) was added at vigorous stirring. The reaction mixture was stirred for 3 h until aldehyde disappeared (TLC control). Next, 1,2-ethylenediamine (10 mL, 150 mmol), CuCl (1 g, 10 mmol) were added and stirred for 1–2 min. After that CF_3_CCl_3_ (18 mL, 150 mmol) was added in one portion at cooling by cold water bath. The reaction mixture stirred overnight at room temperature, poured into HCl water solution (1000 mL, ~0.4–0.5 M) and extracted with CH_2_Cl_2_ (3 × 150 mL). Combined extract was washed with water (200 mL) and dried over Na_2_SO_4_. Solvents were evaporated in vacuo, the residue was purified by passing through a short silica gel pad (~120–150 cm^3^ of silica gel) using 3:1 mixture of hexane as an eluent. Evaporation of the solvents afforded pure **2a** as slightly yellow oil. Yield 17.1 g (68%).

**Synthesis of styrenes 2 by Wittig reaction (general procedure III, 5 mmol scale) [109].** One neck 20 mL vial with a screw cap was charged with corresponding benzaldehyde (2 mmol), PPh_3_ (1.258 g, 4.8 mmol), K_2_CO_3_ (0.028 g, 0.2 mmol), MeCN (2 mL) and CF_3_CCl_3_ (0.561 g, 3 mmol). The reaction mixture was stirred for 3–5 h at 80 °C and then poured into water (100 mL) and extracted with CH_2_Cl_2_ (3 × 20 mL). Combined extract was washed with water (20 mL) and dried over Na_2_SO_4_. Solvents were evaporated in vacuo, the residue was purified by column chromatography on silica gel using 3:1 (**2b,g,i,k,n**) and 1:1 (**2j,l,m**) mixtures of hexane and CH_2_Cl_2_ as eluents.

**1-(2-Chloro-3,3,3-trifluoroprop-1-en-1-yl)-2-nitrobenzene (2a).** Obtained from 2-**nitrobenzaldehyde**. Obtained from 2-nitrobenzaldehyde **1a** (0.151 g, 1 mmol) by procedure II. Сolorless oil, yield 0.223 g (88%). Mixture of *Z*/*E* isomers (82:18; by ^19^F NMR). NMR data of styrene **2a** (see Appendix A) are in agreement with those in the literature [110].

**4-Chloro-1-(2-chloro-3,3,3-trifluoroprop-1-en-1-yl)-2-nitrobenzene (2b).** Obtained from 4-chloro-2-nitrobenzaldehyde **1b** (0.185g, 1 mmol) by procedure II. Light yellow oil, yield 0.223 g (78%). Mixture of *Z*/*E* isomers (90:10; by ^19^F NMR). For the mixture of isomers: *Z*-isomer: ^1^H NMR (CDCl_3_, 400.1 MHz): δ 8.21 (d, 1H, ^4^*J* = 2.1 Hz), 7.75–7.67 (m, 2H), 7.61 (d, 1H, ^3^*J* = 8.3 Hz). ^13^C{^1^H} NMR (CDCl_3_, 100.6 MHz): δ 147.6, 136.4, 133.9, 132.4, 128.2 (q, ^3^*J*_CF_ = 4.8 Hz), 125.9, 125.4, 123.4 (q, ^2^*J*_CF_ = 38.0 Hz), 120.2 (q, ^1^*J*_CF_ = 272.8 Hz). ^19^F NMR (CDCl_3_, 376.5 MHz): δ −70.4 (d, 3F, ^4^*J* = 1.0 Hz). *E*- isomer: ^1^H NMR (CDCl_3_, 400.1 MHz): δ 7.65 (dd, 1H, ^3^*J* = 8.3 Hz, ^4^*J* = 2.2 Hz), 7.46 (s, 1H), 7.32 (d, 1H, ^3^*J* = 8.3 Hz). Other signals are overlapped with those of major isomer. ^13^C{^1^H} NMR (CDCl_3_, 100.6 MHz): δ 136.2, 133.8, 132.1, 126.8, 125.2. Other signals are overlapped with those of major isomer or cannot be seen in the spectrum due to the low concentration of minor isomer. ^19^F NMR (CDCl_3_, 376.5 MHz): δ −63.2 (s, 3F). HRMS (ESI-TOF): *m*/*z* [M + Ag]^+^ Calcd for C_9_H_4_Cl_2_F_3_NO_2_Ag^+^: 393.8610; found: 393.8619.

**1-(2-Chloro-3,3,3-trifluoroprop-1-en-1-yl)-3-methoxy-2-nitrobenzene****(2c).** Obtained from 3-methoxy-2-nitrobenzaldehyde **1c** (0.188 g, 1.039 mmol) by procedure II. Yellow crystals, mp 42–44 °С, yield 0.211 g (75%). Mixture of *Z*/*E* isomers (76:24; by ^19^F NMR). For the mixture of isomers: *Z*-isomer: ^1^H NMR (CDCl_3_, 400.1 MHz): δ 7.50 (t, 1Н, ^3^*J* = 8.2 Hz), 7.34 (d, 1Н, ^3^*J* = 7.8 Hz), 7.24 (s, 1H), 7.13 (d, 1Н, ^3^*J* = 8.5 Hz), 3.91 (s, 3H). ^13^C{^1^H} NMR (CDCl_3_, 100.6 MHz): δ 151.26, 140.4, 131.43, 125.4, 125.0 (q, ^3^*J*_CF_ = 4.5 Hz), 124.9 (q, ^2^*J*_CF_ = 37.6 Hz), 121.0, 120.1 (q, ^1^*J*_CF_ = 273.9 Hz), 114.0, 56.51. ^19^F NMR (CDCl_3_, 376.5 MHz): δ −70.3 (s, 3F). *E*- isomer: ^1^H NMR (CDCl_3_, 400.1 MHz): δ 7.43 (t, 1Н, ^3^*J* = 8.2 Hz), 7.17 (s, 1H), 7.08 (d, 1Н, ^3^*J* = 8.5 Hz), 6.86 (d, 1Н, ^3^*J* = 7.8 Hz), 3.90 (s, 3H). ^13^C{^1^H} NMR (CDCl_3_, 100.6 MHz): δ 151.24, 139.2, 131.48, 130.0 (q, ^3^*J*_CF_ = 2.1 Hz), 126.8, 125.3 (q, ^2^*J*_CF_ = 37.6 Hz), 120.8 (q, ^4^*J*_CF_ = 2.5 Hz), 119.8 (q, ^1^*J*_CF_ = 273.9 Hz), 113.4, 56.46. ^19^F NMR (CDCl_3_, 376.5 MHz): δ −63.6 (s, 3F). HRMS (ESI-TOF): *m*/*z* [M + NH_4_]^+^ Calcd for C_10_H_11_ClF_3_N_2_O_3_^+^: 299.0405; found: 299.0404.

**2-(2-Chloro-3,3,3-trifluoroprop-1-en-1-yl)-4-methoxy-1-nitrobenzene****(2d).** Obtained from 5-methoxy-2-nitrobenzaldehyde **1d** (0.183 g, 1.011 mmol) by procedure II. Yellow oil, yield 0.234 g (82%). Mixture of *Z*/*E* isomers (84:16; by ^19^F NMR). For the mixture of isomers: *Z*-isomer: ^1^H NMR (CDCl_3_, 400.1 MHz): δ 8.25–8.21 (m, 1H), 7.75 (s, 1H), 7.05–6.99 (m, 2H), 3.92 (s, 3H). ^13^C{^1^H} NMR (CDCl_3_, 100.6 MHz): δ 163.6, 140.0, 130.1 (q, ^3^*J*_CF_ = 4.8 Hz), 130.0, 127.7, 120.4 (q, ^1^*J*_CF_ = 272.6 Hz), 121.9 (q, ^2^*J*_CF_ = 37.7 Hz), 116.2, 114.7, 56.1. ^19^F NMR (CDCl_3_, 376.5 MHz): δ −70.2 (s, 3F). *E*- isomer: ^1^H NMR (CDCl_3_, 400.1 MHz): δ 8.20 (d, 1H, ^3^*J* = 6.6 Hz), 7.51 (s, 1H), 6.97 (d, 1H, ^3^*J* = 2.8 Hz), 6.77 (d, 1H, ^3^*J* = 2.7 Hz), 3.90 (s, 3H). ^13^C{^1^H} NMR (CDCl_3_, 100.6 MHz): δ 163.5, 139.2, 134.1 (q, ^3^*J*_CF_ = 2.3 Hz), 130.9, 127.5, 120.1 (q, ^1^*J*_CF_ = 274.2 Hz), 121.2 (q, ^2^*J*_CF_ = 37.5 Hz), 115.9 (q, ^3^*J*_CF_ = 2.6 Hz), 114.5, 56.04. ^19^F NMR (CDCl_3_, 376.5 MHz): δ −63.0 (s, 3F). HRMS (ESI-TOF): *m*/*z* [M + Na]^+^ Calcd for C_10_H_7_ClF_3_NO_3_Na^+^: 303.9959; found: 303.9957.

**1-(2-Chloro-3,3,3-trifluoroprop-1-en-1-yl)-3,5-dimethyl-2-nitrobenzene (2e).** Obtained from 3,5-dimethyl-2-nitrobenzaldehyde **1e** (0.174 g, 0.972 mmol) by procedure II. Yellow oil, yield 0.214 g (77%). Mixture of *Z*/*E* isomers (78:22; by ^19^F NMR). For the mixture of isomers: *Z*-isomer: ^1^H NMR (CDCl_3_, 400.1 MHz): δ 7.36 (s, 1H), 7.31 (s, 1H), 7.17 (s, 1H), 2.40 (s, 3H), 2.36 (s, 3H). ^13^C{^1^H} NMR (CDCl_3_, 100.6 MHz): δ 147.8, 141.5, 133.4, 131.4, 128.1, 126.8 (q, ^3^*J*_CF_ = 4.6 Hz), 124.9, 123.7 (q, ^2^*J*_CF_ = 37.5 Hz), 120.2 (q, ^1^*J*_CF_ = 272.7 Hz), 21.1, 18.2. ^19^F NMR (CDCl_3_, 376.5 MHz): δ −70.3 (s, 3F). *E*- isomer: ^1^H NMR (CDCl_3_, 400.1 MHz): δ 7.23 (s, 1H), 7.13 (s, 1H), 6.94 (s, 1H). Other signals are overlapped with those of major isomer. ^13^C{^1^H} NMR (CDCl_3_, 100.6 MHz): δ 146.6, 141.6, 133.1, 131.8 (q, ^3^*J*_CF_ = 2.3 Hz), 131.6, 127.9 (q, ^4^*J*_CF_ = 2.4 Hz), 126.4, 123.8 (q, ^2^*J*_CF_ = 37.4 Hz), 119.9 (q, ^1^*J*_CF_ = 274.5 Hz), 20.9, 18.3. ^19^F NMR (CDCl_3_, 376.5 MHz): δ −63.5 (s, 3F). HRMS (ESI-TOF): *m*/*z* [M + H]^+^ Calcd for C_11_H_10_ClF_3_NO_2_^+^: 280.0347; found: 280.0641.

**6-(2-Chloro-3,3,3-trifluoroprop-1-en-1-yl)-7-nitro-2,3-dihydrobenzo[b][1,4]****dioxine (2f).** Obtained from 7-nitro-2,3-dihydrobenzo[*b*][1,4]dioxine-6-carbaldehyde **1f** (0.212 g, 1.014 mmol) by procedure II. Pale yellow crystals, mp 104–106 °С, yield 0.241 g (78%). Mixture of *Z*/*E* isomers (80:20; by ^19^F NMR). For the mixture of isomers: *Z*-isomer: ^1^H NMR (CDCl_3_, 400.1 MHz): δ 7.81 (s, 1H), 7.68 (*pseudo-*d, 1H, ^4^*J* = 0.8 Hz), 7.10 (s, 1H), 4.45–4.28 (m, 4H). ^13^C{^1^H} NMR (CDCl_3_, 100.6 MHz): δ 148.2, 143.9, 129.2 (q, ^3^*J*_CF_ = 4.6 Hz), 121.5, 121.4 (q, ^2^*J*_CF_ = 37.5 Hz), 120.5 (q, ^1^*J*_CF_ = 272.3 Hz), 119.3, 115.0, 64.7, 64.3. ^19^F NMR (CDCl_3_, 376.5 MHz): δ −70.2 (d, 3F, ^4^*J* = 1.0 Hz). *E*- isomer: ^1^H NMR (CDCl_3_, 400.1 MHz): δ 7.80 (s,1H), 7.43 (*pseudo-*d, 1H, ^4^*J* = 0.8 Hz), 6.79 (s, 1H). Other signals are overlapped with those of major isomer. ^13^C{^1^H} NMR (CDCl_3_, 100.6 MHz): δ 143.8, 140.5, 133.8 (q, ^3^*J*_CF_ = 2.3 Hz), 122.6, 121.5 (q, ^2^*J*_CF_ = 37.3 Hz), 120.2 (q, ^1^*J*_CF_ = 274.1 Hz), 119.0 (q, ^3^*J*_CF_ = 2.5 Hz), 114.8, 64.2. Other signals are overlapped with those of major isomer or cannot be seen in the spectrum due to the low concentration of minor isomer. ^19^F NMR (CDCl_3_, 376.5 MHz): δ −63.1 (s, 3F). HRMS (ESI-TOF): *m*/*z* [M + H]^+^ Calcd for C_11_H_8_ClF_3_NO_4_^+^: 310.0088; found: 310.0086.

**2-(2-Chloro-3,3,3-trifluoroprop-1-en-1-yl)-1,4-dimethoxy-3-nitrobenzene****(2g).** Obtained from 1,4-dimethoxy-3-nitrobenzaldehyde **1g** (0.222 g, 1.052 mmol) by procedure II. Pale yellow crystals, mp 72–73 °С, yield 0.254 g (78%). Mixture of *Z*/*E* isomers (91:9; by ^19^F NMR). For the mixture of isomers: *Z*-isomer: ^1^H NMR (CDCl_3_, 400.1 MHz): δ 7.18 (s, 1H), 7.07 (d, 1H, ^3^*J* = 9.3 Hz), 7.03 (d, 1H, ^3^*J* = 9.2 Hz), 3.87 (s, 3H), 3.83 (s, 3H). ^13^C{^1^H} NMR (CDCl_3_, 100.6 MHz): δ 150.2, 145.1, 126.8 (q, ^2^*J*_CF_ = 37.8 Hz), 124.0 (q, ^3^*J*_CF_ = 4.6 Hz), 119.9 (q, ^1^*J*_CF_ = 272.9 Hz), 115.4, 114.4, 113.9, 57.0, 56.5. ^19^F NMR (CDCl_3_, 376.5 MHz): δ −70.3 (d, 3F, ^4^*J* = 1.0 Hz). *E*- isomer: ^1^H NMR (CDCl_3_, 400.1 MHz): δ 6.99 (d, 1H, ^3^*J* = 3.2 Hz), 6.98 (d, 1H, ^3^*J* = 3.2 Hz), 6.88 (s, 1H), 3.85 (s, 3H), 3.79 (s, 3H). ^13^C{^1^H} NMR (CDCl_3_, 100.6 MHz): δ 144.7, 140.4, 113.3, 56.9, 56.3. Other signals are overlapped with those of major isomer or cannot be seen in the spectrum due to the low concentration of minor isomer. ^19^F NMR (CDCl_3_, 376.5 MHz): δ −67.8 (s, 3F). HRMS (ESI-TOF): *m*/*z* [M + H]^+^ Calcd for C_11_H_10_ClF_3_NO_4_^+^: 312.0245; found: 312.0251.

**1-(2-Chloro-3,3,3-trifluoroprop-1-en-1-yl)-4,5-dimethoxy-2-nitrobenzene****(2h).** Obtained from 2,5-dimethoxy-3-nitrobenzaldehyde **1h** by procedure I (0.539 g, 2.55 mmol, DMSO) and by procedure II (0.245 g, 1.161 mmol). Pale yellow solid, mp 95–97 °С, yield 0.374 g (47%, I) yield 0.128 g (43%, II). Mixture of *Z*/*E* isomers (80:20; by ^19^F NMR). For the mixture of isomers: *Z*-isomer: ^1^H NMR (CDCl_3_, 400.1 MHz): δ 7.79–7.74 (m, 2H), 7.02 (s, 1H), 3.99 (s, 3H), 3.99 (s, 3H). ^13^C{^1^H} NMR (CDCl_3_, 100.6 MHz): δ 153.2, 149.4, 140.1, 129.9 (q, ^3^*J*_CF_ = 4.7 Hz), 121.6, 121.4 (q, ^2^*J*_CF_ = 37.3 Hz), 120.4 (q, ^1^*J*_CF_ = 272.4 Hz), 112.1, 107.7, 56.6, 56.40. ^19^F NMR (CDCl_3_, 376.5 MHz): δ −70.1 (d, 3F, ^4^*J* = 1.0 Hz). *E*- isomer: ^1^H NMR (CDCl_3_, 400.1 MHz): δ 7.52 (*pseudo-*d, 1H, ^4^*J* = 0.6 Hz), 6.71 (s, 1H), 3.97 (s, 3H), 3.95 (s, 3H). Other signals are overlapped with those of major isomer. ^13^C{^1^H} NMR (CDCl_3_, 100.6 MHz): δ 149.3, 138.9, 134.3 (q, ^3^*J*_CF_ = 2.3 Hz), 122.7, 120.2 (q, ^1^*J*_CF_ = 274.3 Hz), 121.1 (q, ^2^*J*_CF_ = 37.2 Hz), 112.1, 107.5, 56.5, 56.38. ^19^F NMR (CDCl_3_, 376.5 MHz): δ −62.8 (s, 3F). HRMS (ESI-TOF): *m*/*z* [M + H]^+^ Calcd for C_11_H_10_ClF_3_NO_4_^+^: 312.0245; found: 312.0254.

**5-(2-Chloro-3,3,3-trifluoroprop-1-en-1-yl)-6-nitrobenzo[d][1,3]****dioxole (2i).** Obtained from 4,5-ethylendioxy-2-nitrobenzaldehyde **1i** (0.207 g, 1.062 mmol) by procedure II. Pale yellow solid, mp 100–103 °С, yield 0.155 g (52%). Mixture of *Z*/*E* isomers (79:21; by ^19^F NMR). For the mixture of isomers: *Z*-isomer: ^1^H NMR (CDCl_3_, 400.1 MHz): δ 7.68–7.66 (m, 2H), 6.99 (s, 1H). 6.19 (s, 2H). ^13^C{^1^H} NMR (CDCl_3_, 100.6 MHz): δ 152.22, 148.9, 141.8, 129.7 (q, ^3^*J*_CF_ = 4.8 Hz), 123.9, 121.8 (q, ^2^*J*_CF_ = 37.5 Hz), 120.4 (q, ^1^*J*_CF_ = 272.5 Hz), 109.6, 105.7, 103.64. ^19^F NMR (CDCl_3_, 376.5 MHz): δ −70.2 (s, 3F). *E*- isomer: ^1^H NMR (CDCl_3_, 400.1 MHz): δ 7.65 (s, 1H), 7.43 (d, 1Н, ^4^*J* = 0.6 Hz), 6.70 (s, 1H), 6.17 (s, 2H). ^13^C{^1^H} NMR (CDCl_3_, 100.6 MHz): δ 152.19, 148.8, 140.6, 134.0 (q, ^3^*J*_CF_ = 2.8 Hz), 124.9, 121.3 (q, ^2^*J*_CF_ = 37.4 Hz), 120.1 (q, ^1^*J*_CF_ = 274.5 Hz), 109.5 (q, ^4^*J*_CF_ = 2.8 Hz), 105.4, 103.62. ^19^F NMR (CDCl_3_, 376.5 MHz): δ −63.2 (s, 3F). HRMS (ESI-TOF): *m*/*z* [M + Na]^+^ Calcd for C_10_H_5_ClF_3_NO_4_Na^+^: 317.9751; found: 317.9752.

**1-(2-Chloro-3,3,3-trifluoroprop-1-en-1-yl)-2,4-dinitrobenzene (2j).** Obtained from 2,4-dinitrobenzaldehyde by procedure (II, 0.196 g) and (III, 0.65 g). Yellow viscous oil, yield 0.014 g (5%, II), 0.248 (25%, (III). Mixture of *Z*/*E* isomers (96:4; by ^19^F NMR). For the mixture of isomers: *Z*-isomer: ^1^H NMR (CDCl_3_, 400.1 MHz): δ 9.03 (*pseudo-*d, 1H, ^4^*J* ~ 1.5 Hz), 8.58 (dd, 1H, ^3^*J* = 8.5 Hz, ^4^*J* = 2.3 Hz), 7.91 (d, 1H, ^3^*J* = 8.5 Hz), 7.78 (s, 1H). ^13^C{^1^H} NMR (CDCl_3_, 100.6 MHz): δ 148.1, 147.4, 140.2, 133.0, 127.9, 127.6 (q, ^3^*J*_CF_ = 4.5 Hz), 125.0 (q, ^2^*J*_CF_ = 38.0 Hz), 120.6, 119.9 (q, ^1^*J*_CF_ = 273.2 Hz). ^19^F NMR (CDCl_3_, 376.5 MHz): δ −69.4 (d, 3F, ^4^*J* = 0.6 Hz). *E*- isomer: ^1^H NMR (CDCl_3_, 400.1 MHz): δ 7.78 (br.s, 1H), 8.52 (dd, 1H, ^3^*J* = 8.5 Hz, ^4^*J* = 2.3 Hz), 7.64 (d, 1H, ^3^*J* = 8.5 Hz), 7.53 (s, 1H). ^13^C{^1^H} NMR (CDCl_3_, 100.6 MHz): 127.0. Other signals are overlapped with those of major isomer or cannot be seen in the spectrum due to the low concentration of minor isomer. ^19F NMR (CDCl^_3_, 376.5 MHz): δ −62.2 (s, 3F). HRMS (ESI-TOF): *m*/*z* [M-H]^-^ Calcd for C_9_H_3_ClF_3_N_2_O_4_^−^: 294.9739; found: 294.9732.

**1-(2-Chloro-3,3,3-trifluoroprop-1-en-1-yl)-2-nitro-4-(trifluoromethyl)benzene (2k).** Obtained from 2-nitro-4-(trifluoromethyl)benzaldehyde **1k** by procedure II (0.438 g, 2 mmol) and by procedure III (0.438 g, 2 mmol). Yellow oil, yield 0.395 g (62%, II), 0.365 g (57%, III). Mixture of *Z*/*E* isomers (92:8; by ^19^F NMR). For the mixture of isomers: *Z*-isomer: ^1^H NMR (CDCl_3_, 400.1 MHz): δ 8.49 (*pseudo-*d, 1H, ^4^*J* ~ 1.0 Hz), 7.99 (dd, 1H, ^3^*J* = 8.0 Hz, ^4^*J* = 0.7 Hz), 7.81 (d, 1H, ^3^*J* = 8.1 Hz), 7.77 (s, 1H). ^13^C{^1^H} NMR (CDCl_3_, 100.6 MHz): δ 147.2, 132.8 (q, ^2^*J*_CF_ = 34.6 Hz), 132.4, 131.12, 130.3 (q, ^3^*J*_CF_ = 3.4 Hz), 128.2 (q, ^3^*J*_CF_ = 4.8 Hz), 124.3 (q, ^2^*J*_CF_ = 38.1 Hz), 122.53 (q, ^3^*J*_CF_ = 3.8 Hz), 122.46 (q, ^1^*J*_CF_ = 273.1 Hz), 120.1 (q, ^1^*J*_CF_ = 272.8 Hz). ^19^F NMR (CDCl_3_, 376.5 MHz): δ −69.5 (s, 3F), −63.3 (s, 3F). *E*- isomer: ^1^H NMR (CDCl_3_, 400.1 MHz): δ 7.94 (dd, 1H, ^3^*J* = 8.0 Hz, ^4^*J* = 0.7 Hz), 7.56 (s, 1H), 7.53 (d, 1H, ^3^*J* = 8.2 Hz). Other signals are overlapped with those of major isomer. ^19^F NMR (CDCl_3_, 376.5 MHz): δ −62.3 (s, 3F). Other signals are overlapped with those of major isomer. HRMS (ESI-TOF): *m*/*z* [M + Ag]^+^ Calcd for C_10_H_4_ClF_6_NO_2_Ag^+^: 425.8880; found: 425.8874.

**4-(2-Chloro-3,3,3-trifluoroprop-1-en-1-yl)-3-nitrobenzonitrile (2l).** Obtained from 4-cyano-2-nitrobenzaldehyde **1l** by procedure II (0.176 g, 1 mmol) and by procedure III (0.88 g, 5 mmol). Yellow oil, yield 0.070 g (25%) (II), 0.278 g (20%, III). Mixture of *Z*/*E* isomers (96:4; by ^19^F NMR). For the mixture of isomers: *Z*-isomer: ^1^H NMR (CDCl_3_, 400.1 MHz): δ

δ 8.51 (d, 1H, ^4^*J* = 1.6 Hz), 8.01 (dd, 1H, ^3^*J* = 8.1 Hz, ^4^*J* = 1.6 Hz), 7.81 (d, 1H, ^3^*J* = 8.1 Hz), 7.75 (s, 1H). ^13^C{^1^H} NMR (CDCl_3_, 100.6 MHz): δ 147.1, 136.6, 132.5, 131.7, 128.7, 127.9 (q, ^3^*J*_CF_ = 4.7 Hz), 124.3 (q, ^2^*J*_CF_ = 38.5 Hz), 119.8 (q, ^1^*J*_CF_ = 273.0 Hz, CF_3_), 115.9, 114.4. ^19^F NMR (CDCl_3_, 376.5 MHz): δ −70.5 (s, 3F). *E*-isomer: ^1^H NMR (CDCl_3_, 400.1 MHz): δ 7.95 (dd, 1H, ^3^*J* = 8.0 Hz, ^4^*J* = 1.6 Hz), 7.50 (s, 1H). Other signals are overlapped with those of major isomer. ^19^F NMR (CDCl_3_, 376.5 MHz): δ −63.3 (s, 3F). HRMS (ESI-TOF): *m*/*z* [M + H]^+^ Calcd for C_10_H_5_ClF_3_N_2_O_2_^+^: 276.9986; found: 276.9986.

**Methyl 4-(2-chloro-3,3,3-trifluoroprop-1-en-1-yl)-3-nitrobenzoate (2m).** Obtained from methyl 4-formyl-3-nitrobenzoate **1m** by procedure II (0.209 g, 1 mmol) and by procedure III (0.209 g, 1 mmol). Beige crystals, yield 0.040 g (13%, II), 0.079 g (22%, III). Mixture of *Z*/*E* isomers (95:5; by ^19^F NMR). ^1^H NMR (CDCl_3_, 400.1 MHz): 

δ 8.81 (d, 1H, ^4^*J* = 1.7 Hz), 8.35 (dd, 1H, ^3^*J* = 8.0 Hz, ^4^*J* = 1.7 Hz), 7.76 (s, 1H), 7.73 (d, 1H, ^3^*J* = 8.1 Hz), 3.99 (s, 3H). ^13^C{^1^H} NMR (CDCl_3_, 100.6 MHz): δ 164.3, 147.2, 134.2, 132.4, 131.7, 131.4, 128.6 (q, ^3^*J*_CF_ = 4.7 Hz), 126.1, 123.7 (q, ^2^*J*_CF_ = 38.0 Hz), 120.1 (q, ^1^*J*_CF_ = 272.9 Hz), 53.0. ^19^F NMR (CDCl_3_, 376.5 MHz): δ −70.4 (s, 3F). HRMS (ESI-TOF): *m*/*z* [M + H]^+^ Calcd for C_11_H_8_ClF_3_NO_4_^+^: 310.0088; found: 310.0085.

**4-Chloro-2-(2-chloro-3,3,3-trifluoroprop-1-en-1-yl)-1-nitrobenzene (2n).** Obtained from 5-chloro-2-nitrobenzaldehyde **1n** (0.191g, 1.03 mmol) by procedure II. Yellow crystals, mp 46–48 °С, yield 0.221 g (75%). Mixture of *Z*/*E* isomers (91:9; by ^19^F NMR). For the mixture of isomers: *Z*-isomer: ^1^H NMR (CDCl_3_, 400.1 MHz): 

δ 8.19 (d, 1Н, ^3^*J* = 8.8 Hz), 7.70 (s, 1H), 7.61 (d, 1Н, ^4^*J* = 2.1 Hz), 7.58–7.55 (m, 1H). ^13^C{^1^H} NMR (CDCl_3_, 100.6 MHz): δ 145.3, 140.4, 131.1, 130.3, 129.2, 128.3 (q, ^3^*J*_CF_ = 4.8 Hz), 126.5, 123.4 (q, ^2^*J*_CF_ = 38.0 Hz), 120.1 (q, ^1^*J*_CF_ = 272.9 Hz). ^19^F NMR (CDCl_3_, 376.5 MHz): δ −70.5 (s, 3F). *E*- isomer: ^1^H NMR (CDCl_3_, 400.1 MHz): 

δ 7.46 (s, 1H). Other signals are overlapped with those of major isomer. ^19^F NMR (CDCl_3_, 376.5 MHz): δ −63.3 (s, 3F). HRMS (ESI-TOF): *m*/*z* [M + Ag]^+^ Calcd for C_9_H_4_Cl_2_F_3_NO_2_Ag^+^: 395.8583; found: 395.8587.

**2-(2-Chloro-3,3,3-trifluoroprop-1-en-1-yl)-4-fluoro-1-nitrobenzene (2o).** Obtained from 5-fluoro-2-nitrobenzaldehyde **1o** (0.175g, 1.04 mmol) by procedure II. White crystals, mp 35–38 °С, yield 0.204 g (73%). Mixture of *Z*/*E* isomers (83:17; by ^19^F NMR). For the mixture of isomers: *Z*-isomer: ^1^H NMR (CDCl_3_, 400.1 MHz): 

δ 8.28 (dd, 1Н, ^3^*J* = 9.1 Hz, ^3^*J* = 5.1 Hz), 7.73 (s, 1H), 7.34 (dd, 1Н, ^3^*J* = 8.5 Hz, ^4^*J* = 2.6 Hz), 7.31–7.25 (m, 1H). ^13^C{^1^H} NMR (CDCl_3_, 100.6 MHz): δ 164.9 (d, ^1^*J*_CF_ = 259.2 Hz), 143.3 (d, ^4^*J*_CF_ = 2.5 Hz), 130.6 (d, ^3^*J*_CF_ = 10.0 Hz), 128.5 (qd, ^3^*J*_CF_ = 4.5 Hz, ^4^*J*_CF_ = 0.9 Hz), 128.1 (d, ^3^*J*_CF_ = 10.2 Hz), 123.4 (q, ^2^*J*_CF_ = 37.9 Hz), 120.2 (q, ^1^*J*_CF_ = 272.8 Hz), 118.4 (d, ^2^*J*_CF_ = 25.1 Hz), 117.3 (d, ^2^*J*_CF_ = 23.1 Hz). ^19^F NMR (CDCl_3_, 376.5 MHz): δ −70.5 (s, 3F), −102.55–−102.71 (m, 1F). *E*- isomer: ^1^H NMR (CDCl_3_, 400.1 MHz): 

δ 7.48 (s, 1H), 7.25–7.22 (m, 1H) 7.07 (dd, 1Н, ^3^*J* = 8.2 Hz, ^4^*J* = 2.7 Hz). Other signals are overlapped with those of major isomer. ^13^C{^1^H} NMR (CDCl_3_, 100.6 MHz): δ 164.7 (d, ^1^*J*_CF_ = 259.8 Hz), 142.4, 132.3 (br.s), 131.4 (d, ^3^*J*_CF_ = 9.9 Hz), 127.9 (d, ^3^*J*_CF_ = 10.2 Hz), 122.6 (q, ^2^*J*_CF_ = 37.5 Hz), 120.0 (q, ^1^*J*_CF_ = 274.4 Hz), 116.9, 118.0 (q, ^4^*J*_CF_ = 2.4 Hz). ^19^F NMR (CDCl_3_, 376.5 MHz): δ −63.3 (s, 3F), −102.73–−102.88 (m, 1F). HRMS (ESI-TOF): *m*/*z* [M-F]^+^ Calcd for C_9_H_4_ClF_3_NO_2_^+^: 249.9877; found: 249.9873.

**Synthesis of α-****CF_3_-****β-(2-nitroaryl)enamines by the reaction with pyrrolidine in neat (general procedure)** [78]. A one neck 25 mL round bottomed flask was charged with dry pyrrolidine (8.5 mL, 100 mmol), cooled down to −18 °C and corresponding styrene **2** (10 mmol) was added in one portion with vigorous stirring. The reaction mixture was stirred at room temperature for 1–3 h until all starting styrene was consumed (TLC or NMR monitoring). The excess of pyrrolidine was evaporated in vacuum, the viscous residue was dissolved in CH_2_Cl_2_ (50 mL), washed with water (3 × 50 mL) and dried over Na_2_SO_4_. CH_2_Cl_2_ was removed in vacuo, and the residue was filtered through a short silica gel pad using appropriate mixture 1:1 of hexane and CH_2_Cl_2_.

**1-[(1*Z*)-2-(2-Nitrophenyl)-1-(trifluoromethyl)vinil]pyrrolidine (3a).** Obtained from 1-(2-chloro-3,3,3-trifluoroprop-1-en-1-yl)-2-nitrobenzene **2a** (6.04 g, 24 mmol). Yellow oil, yield 6.733 g (98%). Mixture of *Z*/*E* isomers (86:14; ^19^F NMR). NMR data of enamine **3a** (see Appendix A) are in agreement with those in the literature [78].

**1-[(1*Z*)-2-(3-Methoxy-2-nitrophenyl)-1-(trifluoromethyl)vinil]pyrrolidine (3c).** Obtained from 1-(2-chloro-3,3,3-trifluoroprop-1-en-1-yl)-3-methoxy-2-nitrobenzene **2c** (0.211 g, 0.75 mmol). Orange oil, yield 0.190 g (80%). Mixture of *Z*/*E* isomers (84:16; ^19^F NMR). NMR data of enamine **3с**(see Appendix A) are in agreement with those in the literature [84].

**1-Hydroxy-7-methoxy-2-(pyrrolidin-1-yl)-2-(trifluoromethyl)indolin-3-one (5c).** Obtained from 1-[2-chloro-3,3,3-trifluoro-1-propenyl]-3-methoxy-2-nitrobenzene **2c** as an admixture in the synthesis of enamine **3c**. Orange oil, yield 0.036 g (15%). ^1^H NMR (CDCl_3_, 400.1 MHz): 

δ 7.74 (s, 1H), 7.24–7.28 (m, 1H), 7.01–7.15 (m, 2H), 3.90 (s, 3H), 3.11 (dd, 2Н, ^3^*J* = 7.2 Hz), 2.95 (q, 2Н, ^3^*J* = 6.9 Hz), 1.78 (t, 4Н, ^3^*J* = 6.2 Hz). ^13^C{^1^H} NMR (CDCl_3_, 100.6 MHz): δ 192.4, 152.3, 149.3, 124.9, 122.6, 122.4 (q, ^1^*J*_CF_ = 284.8 Hz), 118.9, 115.6, 86.4 (q, ^2^*J*_CF_ = 28.1 Hz), 55.9, 47.8, 24.4. ^19^F NMR (CDCl_3_, 376.5 MHz): δ −73.6 (s, 3F). HRMS (ESI-TOF): *m*/*z* [M + H]^+^ Calcd for C_14_H_16_F_3_N_2_O_3_^+^: 317.1108; found: 317.1109.

**1-[(1*Z*)-2-(5-methoxy-2-nitrophenyl)-1-(trifluoromethyl)vinil]pyrrolidine (3d).** Obtained from 2-(2-chloro-3,3,3-trifluoroprop-1-en-1-yl)-4-methoxy-1-nitrobenzene **2d** (0.976 g, 3.465 mmol). Orange oil, yield 1.074 g (98%). Mixture of *Z*/*E* isomers (86:14; ^19^F NMR). NMR data of enamine **3d** (see Appendix A) are in agreement with those in the literature [84].

**1-[(1*Z*)-(4-nitro-3-(3,3,3-trifluoro-2-(pyrrolidin-1-yl)prop-1-en-1-yl)phenyl]pyrrolidine (3n).** Obtained from styrenes **2n** (0.286 g, 1 mmol) or from styrene **2o** (0.396 g, 1.469 mmol). Yellow orange solid, mp 145–147 °С, yield 0.255 g (72% from **2n**), 0.468 g (90% from **2o**). Mixture of *Z*/*E* isomers (84:16; ^19^F). For the mixture of isomers: *Z*-isomer: ^1^H NMR (CDCl_3_, 400.1 MHz): 

δ 8.07 (d, 1Н, ^3^*J* = 9.3 Hz), 6.39–6.33 (m, 2Н), 6.20 (d, 1H, ^4^*J* = 2.4 Hz), 3.39–3.30 (m, 4Н), 3.02 (t, 4Н, ^3^*J* = 6.4 Hz), 2.11–2.02 (m, 4Н), 1.69–1.80 (m, 4Н). ^13^C{^1^H} NMR (CDCl_3_, 100.6 MHz): δ 150.2, 135.8, 135.0, 133.9 (q, ^2^*J*_CF_ = 28.7 Hz), 127.7, 121.9 (q, ^1^*J*_CF_ = 277.8 Hz), 112.9, 109.21, 104.1 (q, ^3^*J*_CF_ = 6.8 Hz), 50.4 (d, ^4^*J*_CF_ = 1.1 Hz), 47.7, 25.4, 25.3. ^19^F NMR (CDCl_3_, 376.5 MHz): δ −65.8 (s, 3F). *E*- isomer: ^1^H NMR (CDCl_3_, 400.1 MHz): 

δ 8.08 (d, 1Н, ^3^*J* = 9.3 Hz), 5.97 (s, 1Н), 6.27 (d, 1H, ^4^*J* = 2.4 Hz), 3.24 (t, 4Н, ^3^*J* = 6.5 Hz), 1.97–1.90 (m, 4Н). Other signals are overlapped with those of major isomer. ^13^C{^1^H} NMR (CDCl_3_, 100.6 MHz): δ 135.5 (q, ^2^*J*_CF_ = 27.3 Hz), 127.6, 113.8 (q, *J* = 3.4, CH = CCF_3_), 109.24, 106.2 (q, ^3^*J*_CF_ = 3.4 Hz), 49.30 (d, ^4^*J*_CF_ = 1.1 Hz), 47.6, 24.6. Other signals are overlapped with those of major isomer or cannot be seen in the spectrum due to the low concentration of minor isomer. ^19^F NMR (CDCl_3_, 376.5 MHz): δ −59.2 (s, 3F). HRMS (ESI-TOF): *m*/*z* [M + H]^+^ Calcd for C_17_H_21_F_3_N_3_O_2_^+^: 356.1580; found: 356.1581.

**Synthesis of indoles 4 by the reduction of nitro-substituted enamines 3 (general procedure IV)**. A one neck 25 mL round bottomed flask was charged with enamine 3 (0.5 mmol), glacial acetic acid (2 mL), water (0.2 mL) and Fe powder (0.112 g, 2 mmol). Reaction mixture was kept at 80 °С under stirring for 1–2 h until dissolving of Fe powder. Volatiles were evaporated in vacuo, the residue was suspended in CH_2_Cl_2_ (2–5 mL) and transferred on the short silica gel pad. The product was isolated using appropriate mixture of hexane and CH_2_Cl_2_ (3:1 for **4a**, **4d**); and mixture of CH_2_Cl_2_ and MeOH (100:1 for **4o**) as eluents.

**Multi-gram scale synthesis of indole 4a**. A one neck 250 mL round bottomed flask was charged with enamine **3a** (7.01 g, 24.5 mmol), glacial acetic acid (100 mL), water (20 mL) and Fe powder (5.49 g, 98 mmol). Reaction mixture was kept at 80–90 °С under stirring for 2 h until dissolving of Fe powder. The reaction mixture was poured into water (1000 mL), the precipitate formed was filtered off and washed by water (100 mL). Next, precipitate was washed with CH_2_Cl_2_ (2 × 50 mL), organic phase was dried over Na_2_SO_4_ and evaporated in vacuo to give pure indole **4a** as colorless plates.

**One pot synthesis of indoles 4 from styrenes 2 (general procedure V)**. A one neck 25 mL round bottomed flask was charged with pyrrolidine (1 mL, 11.8 mmol) and corresponding styrene **2** (0.5 mmol) was added in one portion with vigorous stirring. The reaction mixture was stirred at room temperature for 1–3 h until all starting styrene was consumed (TLC or NMR monitoring). The excess of pyrrolidine was evaporated in vacuum and the viscous residue was dissolved in glacial acetic acid (2 mL) and water (0.2 mL). After that Fe powder (0.112 g, 2 mmol) was added and the reaction mixture was kept at 80 °С under stirring for 1–2 h until dissolving of Fe powder. Volatiles were evaporated in vacuo, the residue was suspended in CH_2_Cl_2_ (2–5 mL) and transferred on the short silica gel pad. The product was isolated using appropriate mixtures of hexane and CH_2_Cl_2_ (3:1 for **4b**,**4c**,**4e,4k,4n**; 1:1 for **4f****,4g,4h,4i**); CH_2_Cl_2_ (for **4l,4m**) and mixture of CH_2_Cl_2_ and MeOH (100:1 for **4j,4o**) as eluents.

**2-(Trifluoromethyl)-1*H*-indole (4a)**. Obtained from enamine **3a** (0.107 g, 0.374 mmol) by procedure IV. White crystals, m.p. 111–112 **°**C, yield 0.059 g (85%). NMR data of indole **4a** (see Appendix A) are in agreement with those in the literature [67].

**6-Chloro-2-(trifluoromethyl)-1*H*-indole (4b)**. Obtained from styrene **2b** (0.100 g, 0.35 mmol) by procedure V. Slightly yellow oil, yield 0.035 g (48%). NMR data of indole **4b** (see Appendix A) are in agreement with those in the literature [67].

**7-Methoxy-2-(trifluoromethyl)-1*H*-indole (4c)**. Obtained from styrene **2c** (0.149 g, 0.53 mmol) by procedure V. Colorless oil, yield 0.058 g (51%). NMR data of indole **4c** (see SI) are in agreement with those in the literature [67].

**5-Methoxy-2-(trifluoromethyl)-1*H*-indole (4d)**. Obtained from enamine **3d** (0.088 g, 0.28 mmol) by procedure IV. Colorless crystals, m.p. 48–49 **°**C, yield 0.0382 g (64%). NMR data of indole **4d** (see Appendix A) are in agreement with those in the literature [67].

**5,7-Dimethyl-2-(trifluoromethyl)-1*H*-indole (4e)**. Obtained from styrene **2e** (0.109 g, 0.391 mmol) by procedure V. Slightly yellow oil, yield 0.036 g (43%). ^1^H NMR (CDCl_3_, 400.1 MHz): δ 8.16 (br.s, 1H), 7.30 (s, 1Н), 6.96 (s, 1Н), 6.88–6.82 (m, 1Н), 2.48 (s, 3H), 2.42 (s, 3H). ^13^C{^1^H} NMR (CDCl_3_, 100.6 MHz): δ 134.3, 130.7, 127.0, 126.5, 125.4 (q, ^2^*J*_CF_ = 38.9 Hz), 121.4 (q, ^1^*J*_CF_ = 267.4 Hz), 120.6, 119.0, 104.3 (q, ^3^*J*_CF_ = 3.4 Hz), 21.3, 16.5. ^19^F NMR (CDCl_3_, 376.5 MHz): δ −61.6 (d, 3F, ^4^*J* = 1.0 Hz). HRMS (ESI-TOF): *m*/*z* [M-H]^-^ Calcd for C_11_H_9_F_3_N^−^: 212.0693; found: 212.0690.

**7-(Trifluoromethyl)-2,3-dihydro-6*H*-[1,4]dioxino[2,3-*f*]-indole (4f)**. Obtained from styrene **2f** (0.154 g, 0.497 mmol) by procedure V. White powder, m.p. 136–138 **°**C, yield 0.098 g (81%). ^1^H NMR (CDCl_3_, 400.1 MHz): δ 8.24 (br.s, 1H), 7.13 (s, 1Н), 6.87 (s, 1Н), 6.77 (s, 1Н), 4.28 (q, 4Н, ^3^*J* = 5.2 Hz). ^13^C{^1^H} NMR (CDCl_3_, 100.6 MHz): δ 143.1, 140.1, 131.7, 125.5 (q, ^2^*J*_CF_ = 38.9 Hz), 121.2 (q, ^1^*J*_CF_ = 267.3 Hz), 121.0, 107.9, 103.8 (q, ^3^*J*_CF_ = 3.4 Hz), 98.6, 64.5, 64.1. ^19^F NMR (CDCl_3_, 376.5 MHz): δ −61.5 (s, 3F). HRMS (ESI-TOF): *m*/*z* [M-H]^-^ Calcd for C_11_H_7_F_3_NO_2_^−^: 242.0434; found: 242.0437.

**4,7-Dimethoxy-2-(trifluoromethyl)-1*H*-indole (4g)**. Obtained from styrene **2g** (0.107 g, 0.309 mmol) by procedure V. Light beige crystals, m.p. 74–76 **°**C, yield 0.053 g (70%). ^1^H NMR (CDCl_3_, 400.1 MHz): δ 8.73 (br.s, 1H), 7.05–7.01 (m, 1Н), 6.62 (d, 1Н, ^3^*J* = 8.3 Hz), 6.42 (d, 1Н, ^3^*J* = 8.3 Hz), 3.92 (s, 3H), 3.91 (s, 3H). ^13^C{^1^H} NMR (CDCl_3_, 100.6 MHz): δ 148.2, 140.9, 128.2, 124.3 (q, ^2^*J*_CF_ = 39.5 Hz), 121.2 (q, ^1^*J*_CF_ = 267.5 Hz), 119.0, 103.9, 102.2 (q, ^3^*J*_CF_ = 3.3 Hz), 99.6, 55.7, 55.6. ^19^F NMR (CDCl_3_, 376.5 MHz): δ −61.4 (d, 3F, ^4^*J* = 0.9 Hz). HRMS (ESI-TOF): *m*/*z* [M]^+^ Calcd for C_11_H_10_F_3_NO_2_^+^: 245.0658; found: 245.0667.

**5,6-Dimethoxy-2-(trifluoromethyl)-1*H*-indole (4h)**. Obtained from styrene **2h** (0.129 g, 0.416 mmol) by procedure V. White crystals, m.p. 89–90 **°**C, yield 0.055 g (54%). NMR data of indole **4h** (see Appendix A) are in agreement with those in the literature [67].

**6-(Trifluoromethyl)-5*H*-[1,3]dioxolo[4.5-*f*]-indole (4i)**. Obtained from styrene **2i** (0.125 g, 0.38 mmol) by procedure V. White crystals, m.p. 113–115 **°**C, yield 0.022 g (25%). NMR data of indole **4i** (see Appendix A) are in agreement with those in the literature [74].

**2-(Trifluoromethyl)-1*H*-indole-6-amine (4j)**. Obtained from styrene **2j** (0.293 g, 0.99 mmol) by procedure V. 8 Equivalents of Fe (0.448 g, 8 mmol) was used due to the presence of second nitro-group in the styrene **2j**. Beige crystals, m.p. 124–126 **°**C, yield 0.119 g (60%). NMR data of indole **4j** (see Appendix A) are in agreement with those in the literature [67].

**2,6-Bis(trifluoromethyl)-1*H*-indole (4k)**. Obtained from styrene **2k** (0.240 g, 0.75 mmol) by procedure V. Yellow crystals, m.p. 46–47 **°**C, yield 0.0896 g (47%). NMR data of indole **4k** (see Appendix A) are in agreement with those in the literature [67].

**2-(Trifluoromethyl)-1*H*-indole-6-carbonitril (4l)**. Obtained from styrene **2l** (0.080 g, 0.291 mmol) by procedure V. Slightly brown solid, m.p. 112–114 **°**C, yield 0.0305 g (50%). ^1^H NMR (CDCl_3_, 400.1 MHz): δ 9.18 (br.s, 1H), 7.85 (*pseudo-*d, 1Н, ^4^*J* ~ 1.1 Hz), 7.77 (d, 1Н, ^3^*J* = 8.3 Hz), 7.43 (dd, 1Н, ^3^*J* = 8.3 Hz, ^4^*J* = 1.3 Hz), 6.99 (*pseudo-*dt, 1Н, ^4^*J* ~ 2.1 Hz, ^4^*J* ~ 1.0 Hz). ^13^C{^1^H} NMR (CDCl_3_, 100.6 MHz): δ 134.9, 129.7, 129.5 (q, ^2^*J*_CF_ = 39.2 Hz), 123.7, 123.1, 120.6 (q, ^1^*J*_CF_ = 268.6 Hz), 119.8, 117.0, 107.2, 104.4 (q, ^3^*J*_CF_ = 3.2 Hz). ^19^F NMR (CDCl_3_, 376.5 MHz): δ −62.2 (d, 3F, ^4^*J* = 0.9 Hz). HRMS (ESI-TOF): *m*/*z* [M-H]^-^ Calcd for C_10_H_4_F_3_N_2_^−^: 209.0332; found: 209.0323.

**Methyl 2-(trifluoromethyl)-1*H*-indole-6-carboxylate (4m)**. Obtained from styrene **2m** (0.126 g, 0.408 mmol) by procedure V. Pale brown solid, yield 0.0525 g (53%). NMR data of indole **4m** (see Appendix A) are in agreement with those in the literature [67].

**5-Cloro-2-(trifluoromethyl)-1*H*-indole (4n)**. Obtained from styrene **2n** (0.083 g, 0.29 mmol) by procedure V (piperidine was used instead of pyrrolidine). Pale yellow crystals, m.p. 59–61 **°**C, yield 0.0327 g (71%). NMR data of indole **4n** (see Appendix A) are in agreement with those in the literature [64].

**5-(Pyrrolidin-1-yl)-2-(trifluoromethyl)-1*H*-indole (10a)**. Obtained from enamine **3n** (0.160 g, 0.45 mmol) by procedure V. Orange crystals, m.p. 130–131 **°**C, yield 0.052 g (45%). ^1^H NMR (CDCl_3_, 400.1 MHz): δ 8.11 (br.s, 1H), 7.26 (d, 1Н, ^3^*J* = 9.1 Hz), 6.86–6.70 (m, 3Н), 3.32 (t, 4Н, ^3^*J* = 6.6 Hz), 2.09–2.00 (m, 4H). ^13^C{^1^H} NMR (CDCl_3_, 100.6 MHz): δ 143.9, 129.4, 127.9, 125.6 (q, ^2^*J*_CF_ = 38.5 Hz), 121.4 (q, ^1^*J*_CF_ = 267.5 Hz), 113.2, 112.1, 103.2 (q, ^3^*J*_CF_ = 3.3 Hz), 101.7, 48.6, 25.3. ^19^F NMR (CDCl_3_, 376.5 MHz): δ −61.5 (s, 3F). HRMS (ESI-TOF): *m*/*z* [M + H]^+^ Calcd for C_13_H_14_F_3_N_2_^+^: 255.1104; found: 255.1109.

**One pot synthesis of indoles 10 from styrenes 2 (general procedure VI)**. A 4 mL vial with a screw cup was charged with corresponding amine (5 mmol) and styrene **2n** (0.5 mmol). The reaction mixture was heated at appropriate temperature for several hours (see further) or at room temperature (for MeNH_2_) until starting styrene was consumed (TLC or NMR monitoring). The excess of amine was evaporated in vacuo, the viscous residue was dissolved in glacial acetic acid (2 mL) and transferred into a one neck 25 mL round bottomed flask. Next, water (0.2 mL), Fe powder (0.112 g, 2 mmol) was added, and the reaction mixture was kept at 80 °С at stirring for 1–2 h until dissolving of Fe powder. Volatiles were evaporated in vacuo, the residue was suspended in CH_2_Cl_2_ (2–5 mL) and filtered through a short celite pad. The filtrate was evaporated, and the residue was purified by column chromatography on silica gel using appropriate mixtures of CH_2_Cl_2_ and MeOH (100:1 for **10b-e** and 30:1 for **10f****,g**) as eluents.

**5-(Piperidin-1-yl)-2-(trifluoromethyl)-1*H*-indole (10b)**. Obtained styrene **2n** (0.109 g, 0.404 mmol) and piperidine (0.572 g) by heating at 90 **°**C for 3 h. Pale green-brown solid, m.p. 104–106 **°**C, yield 0.048 g (44%). ^1^H NMR (CDCl_3_, 400.1 MHz): δ 8.46 (br.s, 1H), 7.25 (d, 1Н, ^3^*J* = 8.9 Hz), 7.17 (*pseudo-*d, 1Н, ^4^*J* ~ 2.1 Hz), 7.12 (dd, 1Н, ^3^*J* = 8.9 Hz, ^4^*J* = 2.3 Hz), 6.82 (br.s, 1H), 3.14–3.07 (m, 4Н), 1.77 (dt, 4H, ^3^*J* = 11.3 Hz, ^3^*J* = 5.7 Hz), 1.62–1.54 (m, 2H). ^13^C{^1^H} NMR (CDCl_3_, 100.6 MHz): δ 147.9, 131.5, 127.2, 125.8 (q, ^2^*J*_CF_ = 38.8 Hz), 121.3 (q, ^1^*J*_CF_ = 267.6 Hz), 119.5, 112.1, 108.4, 103.9 (q, ^3^*J*_CF_ = 3.4 Hz), 53.1, 26.2, 24.2. ^19^F NMR (CDCl_3_, 376.5 MHz): δ −61.4 (s, 3F). HRMS (ESI-TOF): *m*/*z* [M + H]^+^ Calcd for C_14_H_16_F_3_N_2_^+^: 269.1260; found: 269.1265.

**4-(2-(Trifluoromethyl)-1*H*-indol-5yl)morpholine (10c)**. Obtained from styrene **2n** (0.104 g, 0.385 mmol) and morpholine (0.530 g) by heating at 100 **°**C for 4 h. Pale green-brown solid, m.p. 167–169 **°**C, yield 0.061 g (59%). ^1^H NMR (CDCl_3_, 400.1 MHz): 

δ 9.91 (br.s, 1Н), 7.42–7.36 (m, 1Н), 7.12–7.07 (m, 2Н), 6.84 (*pseudo*-dt, 1 H, ^4^*J* ~ 2.1 Hz, ^4^*J* ~ 1.0 Hz), 3.84–3.75 (m, 4H), 3.10–3.01 (m, 4H). ^13^C{^1^H} NMR (CDCl_3_, 100.6 MHz): δ 147.8, 133.0, 127.9, 126.3 (q, ^2^*J*_CF_ = 38.6 Hz), 122.6 (q, ^1^*J*_CF_ = 266.5 Hz), 118.9, 113.5, 107.8, 104.1 (q, ^3^*J*_CF_ = 3.4 Hz), 67.6, 52.1. ^19^F NMR (CDCl_3_, 376.5 MHz): δ −59.5 (d, 3F, ^4^*J* = 1.0 Hz). HRMS (ESI-TOF): *m*/*z* [M + H]^+^ Calcd for C_13_H_14_F_3_N_2_O^+^: 271.1053; found: 271.1057.

**5-(Azepan-1-yl)-2-(trifluoromethyl)-1*H*-indole (10d)**. Obtained from styrene **2n** (0.107 g, 0.396 mmol) and hexamethyleneimine (0.480 g) by heating at 100 **°**C for 4 h. Pale yellow-brown solid, m.p. 65–67 **°**C, yield 0.060 g (54%). ^1^H NMR (CDCl_3_, 400.1 MHz): δ 8.12 (br.s, 1Н), 7.23 (d, 1Н, ^3^*J* = 9.0 Hz), 6.92 (dd, 1Н, ^3^*J* = 9.0 Hz, ^4^*J* = 2.4 Hz), 6.88 (*pseudo*-d, 1 H, ^4^*J* ~ 2.2 Hz), 6.79 (br.s, 1H), 3.56–3.47 (m, 4H), 1.89–1.79 (m, 4H), 1.61–1.53 (m, 4H). ^13^C{^1^H} NMR (CDCl_3_, 100.6 MHz): δ 144.6, 129.1, 128.0, 125.6 (q, ^2^*J*_CF_ = 38.7 Hz), 121.4 (q, ^1^*J*_CF_ = 267.4 Hz), 112.9, 112.2, 103.3 (q, ^3^*J*_CF_ = 3.1 Hz), 101.5, 50.0, 27.9, 27.1. ^19^F NMR (CDCl_3_, 376.5 MHz): δ −61.5 (d, 3F, ^4^*J* = 0.9 Hz). HRMS (ESI-TOF): *m*/*z* [M + H]^+^ Calcd for C_15_H_18_F_3_N_2_^+^: 283.1417; found: 283.1424.

***N,N*-Diethyl-2-(trifluoromethyl)-1*H*-indole-5-amine (10e)**. Obtained from styrene **2n** (0.101 g, 0.374 mmol) and diethylamine (0.480 g) by heating at 100 **°**C for 10 h. Pale brown oil, yield 0.041 g (43%). ^1^H NMR (CDCl_3_, 400.1 MHz): δ 8.29 (br.s, 1Н), 7.26 (d, 1Н, ^3^*J* = 8.7 Hz), 7.02–6.93 (m, 2Н), 6.79 (s, 1H), 3.33 (q, 4H, ^3^*J* = 7.1 Hz), 1.13 (t, 6H, ^3^*J* = 7.1 Hz). ^13^C{^1^H} NMR (CDCl_3_, 100.6 MHz): δ 143.7, 130.3, 127.6, 125.7 (q, ^2^*J*_CF_ = 38.9 Hz), 121.4 (q, ^1^*J*_CF_ = 267.4 Hz), 116.4, 112.2, 106.0, 103.5 (q, ^3^*J*_CF_ = 3.2 Hz), 45.9, 12.3. ^19^F NMR (CDCl_3_, 376.5 MHz): δ −61.6 (d, 3F, ^4^*J* = 1.1 Hz). HRMS (ESI-TOF): *m*/*z* [M + H]^+^ Calcd for C_13_H_16_F_3_N_2_^+^: 257.1260; found: 257.1261.

***N*-Methyl-2-(trifluoromethyl)-1*H*-indole-5-amine (10f)**. Obtained from styrene **2n** (0.116 g, 0.430 mmol) and *n*-methylamine (2 mL of 3.65 M solution in MeOH) by keeping the reaction mixture for 11 days. Pale green-brown solid, m.p. 133–135 **°**C, yield 0.040 g (44%). ^1^H NMR (CD_3_CN, 400.1 MHz): δ 9.74 (br.s, 1Н), 7.26 (d, 1Н, ^3^*J* = 8.7 Hz), 6.80–6.68 (m, 3Н), 2.77 (s, 3H). ^13^C{^1^H} NMR (CD_3_CN, 100.6 MHz): δ 145.9, 131.4, 128.5, 125.7 (q, ^2^*J*_CF_ = 38.5 Hz), 122.8 (q, ^1^*J*_CF_ = 266.3 Hz), 116.2, 113.5, 103.4 (q, ^3^*J*_CF_ = 3.4 Hz), 101.0, 31.4. ^19^F NMR (CD_3_CN, 376.5 MHz): δ −59.3 (d, 3F, ^4^*J* = 0.9 Hz). HRMS (ESI-TOF): *m*/*z* [M + H]^+^ Calcd for C_10_H_10_F_3_N_2_^+^: 215.0791; found: 215.0792.

***N*-Hexyl-2-(trifluoromethyl)-1*H*-indole-5-amine (10g)**. Obtained from styrene **2n** (0.100 g, 0.370 mmol) and *n*-hexylamine (0.482 g) by heating at 100 **°**C for 4 h. Pale yellow-brown solid, m.p. 88–90 **°**C, yield 0.047 g (45%). ^1^H NMR (CDCl_3_, 400.1 MHz): δ 8.31 (br.s, 1Н), 7.17 (d, 1Н, ^3^*J* = 8.8 Hz), 6.82 (d, 1Н, ^4^*J* = 2.1 Hz), 6.77–6.69 (m, 2H), 3.16–3.10 (m, 2Н), 2.96 (br.s, 1H), 1.65 (dt, 2Н, ^3^*J* = 14.7 Hz, ^3^*J* = 7.2 Hz), 1.48–1.29 (m, 6H), 0.91 (t, 3H, ^3^*J* = 7.0 Hz). ^13^C{^1^H} NMR (CDCl_3_, 100.6 MHz): δ 143.4, 130.3, 127.7, 125.6 (q, ^2^*J*_CF_ = 38.6 Hz), 121.4 (q, ^1^*J*_CF_ = 267.4 Hz), 115.4, 112.3, 103.3 (q, ^3^*J*_CF_ = 3.3 Hz), 102.2, 45.1, 31.7, 29.5, 26.9, 22.6, 14.0. ^19^F NMR (CDCl_3_, 376.5 MHz): δ −61.5 (d, 3F, ^4^*J* = 1.0 Hz). HRMS (ESI-TOF): *m*/*z* [M + H]^+^ Calcd for C_15_H_20_F_3_N_2_^+^: 285.1573; found: 285.1576.


**Reactions of indole 4a with electrophiles.**


**Synthesis of 2-(trifluoromethyl)-1*H*-indol-3-carbaldehyde (17)**. A 4 mL vial with a screw cup was charged with DMF (0.5 mL), cooled to −18 °C (in the fridge) and then POCl_3_ (0.210 g, 1.37 mmol) was added. The reaction mixture was kept at 5–7 °C (in the fridge) for 30 min and then indole **4a** (0.108 g, 0.58 mmol). The reaction mixture was stirred for 6h at 80 °C, cooled down to room temperature and transferred to separating funnel with water (50 mL) using CH_2_Cl_2_ (30–40 mL). After shaking, organic phase was separated, water phase was extracted with CH_2_Cl_2_ (20 mL). Combined organic phase was washed with water (20 mL), and dried over Na_2_SO_4_. Volatiles were evaporated in vacuo, the residue formed was suspended in hexane-CH_2_Cl_2_ mixture (3:1, 2 mL). The precipitate was filtered off and dried in vacuo to give pure **17**. Beige powder, m.p. 167–169 **°**C, yield 0.066 g (53%). NMR data of indole **17** (see Appendix A) are in agreement with those in the literature [66].

**1-(2-(Trifluoromethyl)-1*H*-indol-3-yl)ethanone (18)**. An 8 mL vial with a screw cup was charged with 1,2-dichloroethane (1.5 mL), AlCl_3_ (0.124 g, 0.93 mmol), cooled to −18 **°**C (in the fridge) and then AcCl (0.047 g, 0.60 mmol) was added. The reaction mixture was stirred at room temperature for 30 min and then indole **4a** (0.089 g, 0.48 mmol) was added. The reaction mixture was stirred overnight and poured into water (50 mL). Water phase was extracted with CH_2_Cl_2_ (3 × 20 mL). Combined organic phase was washed with water (20 mL), and dried over Na_2_SO_4_. Volatiles were evaporated in vacuo, the residue was purified by column chromatography on silica gel using CH_2_Cl_2_ followed by mixture of CH_2_Cl_2_ and MeOH (100:1) as eluents. Beige powder, m.p. 125–127 **°**C, yield 0.070 g (64%). ^1^H NMR (CD_3_CN, 400.1 MHz): δ 10.77 (br.s, 1Н), 8.11 (d, 1Н, ^3^*J* = 8.2 Hz), 7.60–7.56 (m, 1Н), 7.41–7.36 (m, 1Н), 7.35–7.30 (m, 1H), 2.66 (s, 3H). ^13^C{^1^H} NMR (DMSO-*d_6_*, 100.6 MHz): δ 192.7, 134.8, 126.9 (q, ^2^*J*_CF_ = 38.1 Hz), 125.4, 125.3, 124.8 (d, ^4^*J*_CF_ = 3.0 Hz), 123.0, 121.9, 121.1 (q, ^1^*J*_CF_ = 269.6 Hz), 116.9 (q, ^3^*J*_CF_ = 1.5 Hz), 113.4 (d, ^3^*J*_CF_ = 6.2 Hz), 31.0. ^19^F NMR (CD_3_CN, 376.5 MHz): δ −58.0 (s, 3F). HRMS (ESI-TOF): *m*/*z* [M + H]^+^ Calcd for C_11_H_9_F_3_NO^+^: 228.0631; found: 228.0635.

**(*E*)-1,1,1-Trifluoro-4-(2-(trifluoromethyl)-1H-indol-3-yl)but-3-en-2-one (20)**. An 8 mL vial with a screw cup was charged with indole **4a** (0.091 g, 0.49 mmol), (*E*)-4-ethoxy-1,1,1-trifluorobut-3-en-2-one **19** (0.090 g, 0.54 mmol), 1,2-dichloroethane (1 mL), and BF_3_·Et_2_O (0.083 g, 0.059 mmol). The reaction mixture was stirred for 2h at 80 °C and poured into water (30 mL). Water phase was extracted with CH_2_Cl_2_ (3 × 20 mL). Combined organic phase was washed with water (20 mL), and dried over Na_2_SO_4_. Volatiles were evaporated in vacuo, the residue was purified by column chromatography on silica gel using mixtures of hexane and CH_2_Cl_2_ (3:1 followed by 1:1) as eluents. Yellow powder, m.p. 125–127 **°**C, yield 0.0563 g (37%). ^1^H NMR (CDCl_3_, 400.1 MHz): δ 9.07 (br.s, 1Н), 8.30 (d, 1Н, ^3^*J* = 15.9 Hz), 7.98 (d, 1Н, ^3^*J* = 8.0 Hz), 7.53 (d, 1Н, ^3^*J* = 8.0 Hz), 7.50–7.43 (m, 1Н), 7.43–7.38 (m, 1H), 7.20 (d, 1Н, ^3^*J* = 15.9 Hz). ^13^C{^1^H} NMR (CDCl_3_, 100.6 MHz): δ 180.1 (q, ^2^*J*_CF_ = 35.1 Hz), 139.7, 135.3, 128.9 (q, ^2^*J*_CF_ = 37.4 Hz), 126.2, 125.0, 123.7, 121.7, 120.7 (q, ^1^*J*_CF_ = 270.5 Hz), 116.5 (q, ^1^*J*_CF_ = 290.6 Hz), 116.6, 112.8, 112.7 (q, ^3^*J*_CF_ = 2.3 Hz). ^19^F NMR (CDCl_3_, 376.5 MHz): δ −59.0 (d, 3F, ^4^*J* = 0.8 Hz), −78.7 (s, 3F). HRMS (ESI-TOF): *m*/*z* [M + H]^+^ Calcd for C_13_H_8_F_6_NO^+^: 308.0505; found: 308.0509. 

**Reactions of indole 4a with benzaldehydes in alcohols under catalysis with MeSO_3_H (general procedure VII).** A 4 mL vial with a screw cup was charged with indole **4a** (0.0925 g, 0.5 mmol), alcohol (MeOH or EtOH, 1 mL), corresponding benzaldehyde (0.6 mmol or 0.25 mmol for **23**) and MeSO_3_H (0.050g, 0.53 mmol). The reaction mixture was heated at 80 °C for appropriate time, volatiles were evaporated in vacuo, the residue was purified by column chromatography on silica gel using mixtures of hexane and CH_2_Cl_2_ (3:1 followed by 1:1) as eluents.

**3-(Methoxy(phenyl)methyl)-2-(trifluoromethyl)-1*H*-indole (21a)**. Obtained by the reaction of **4a** (0.0925 g, 0.5 mmol) with benzaldehyde (0.065 g, 0.6 mmol) in MeOH by heating for 8h. White crystals, m.p. 86–88 **°**C, yield 0.100 g (68%). NMR data of indole **21a** (see Appendix A) are in agreement with those in the literature [84].

**3-((4-Chlorophenyl)(methoxy)methyl)-2-(trifluoromethyl)-1H-indole (21b)**. Obtained by the reaction of **4a** (0.0925 g, 0.5 mmol) with 4-chlorobenzaldehyde (0.084 g, 0.6 mmol) in MeOH by heating for 10h. White crystals, m.p. 112–113 **°**C, yield 0.112 g (66%). ^1^H NMR (CDCl_3_, 400.1 MHz): δ 8.44 (br.s, 1Н), 7.72 (d, 1Н, ^3^*J* = 8.1 Hz), 7.46–7.35 (m, 3Н), 7.35–7.25 (m, 3Н), 7.11 (ddd, 1Н, ^3^*J* = 8.1 Hz, ^3^*J* = 7.0 Hz, ^4^*J* = 1.0 Hz), 5.79 (s, 1H), 3.41 (s, 3Н). ^13^C{^1^H} NMR (CDCl_3_, 100.6 MHz): δ 139.8, 135.4, 133.0, 128.3, 127.7, 125.2, 125.1, 123.2 (q, ^2^*J*_CF_ = 37.1 Hz), 122.7, 121.7 (q, ^1^*J*_CF_ = 269.3 Hz), 121.2, 117.3 (q, ^3^*J*_CF_ = 2.4 Hz), 111.7, 56.9. ^19^F NMR (CDCl_3_, 376.5 MHz): δ −58.2 (s, 3F). HRMS (ESI-TOF): *m*/*z* [M-MeO]^-^ Calcd for C_16_H_10_ClF_3_N^+^: 308.0448; found: 308.0450.

**3-(Methoxy(4-methoxyphenyl)methyl)-2-(trifluoromethyl)-1*H*-indole (21c)**. Obtained by the reaction of **4a** (0.098 g, 0.53 mmol) with 4-methoxybenzaldehyde (0.087 g, 0.636 mmol) in MeOH by heating for 12h. Pale brown powder, m.p. 138-140 °С, yield 0.092 g (52%). NMR data of indole **21c** (see Appendix A) are in agreement with those in the literature [84].

**3-(Ethoxy(phenyl)methyl)-2-(trifluoromethyl)-1*H*-indole (22)**. Obtained by the reaction of **4a** (0.048 g, 0.259 mmol) with benzaldehyde (0.033 g, 0.306 mmol) in EtOH by heating for 8h. White crystals, m.p. 129–132 **°**C, yield 0.061 g (74%). NMR data of indole **22** (see Appendix A) are in agreement with those in the literature [84].

**3,3’-(Phenylmethylene)bis(2-(trifluoromethyl)-1*H*-indole) (23)**. Obtained by the reaction of **4a** (0.087 g, 0.47 mmol) with benzaldehyde (0.026 g, 0.241 mmol) in EtOH by heating for 12h. Brown oil, yield 0.0486 g (45%). ^1^H NMR (CDCl_3_, 400.1 MHz): δ 8.41 (br.s, 2Н), 7.39 (d, 2Н, ^3^*J* = 8.3 Hz), 7.27 (d, 2Н, ^4^*J* = 2.2 Hz), 7.25–7.16 (m, 5H), 6.84 (ddd, 2Н, ^3^*J* = 8.1 Hz, ^3^*J* = 7.0 Hz, ^4^*J* = 1.0 Hz), 6.72 (d, 2Н, ^3^*J* = 8.1 Hz), 6.54 (s, 1H). ^13^C{^1^H} NMR (CDCl_3_, 100.6 MHz): δ 142.0, 135.0, 128.8, 128.3, 127.2, 126.8, 124.3, 122.4 (q, ^2^*J*_CF_ = 37.5 Hz), 122.3, 121.7 (q, ^1^*J*_CF_ = 269.6 Hz), 120.8, 118.8 (q, ^3^*J*_CF_ = 1.5 Hz), 111,7, 38,0. ^19^F NMR (CDCl_3_, 376.5 MHz): δ −60.0 (s, 3F). HRMS (ESI-TOF): *m*/*z* [M + H]^+^ Calcd for C_25_H_17_F_6_N_2_^+^: 459.1290; found: 459.1290.

## 4. Conclusions

In conclusion, we elaborated a novel three-step pathway towards 2-CF_3_-indoles starting from 2-nitrobenzaldehydes. Catalytic olefination reaction of 2-nitrobenzaldehydes with CF_3_CCl_3_ leads, efficiently, to the corresponding trifluoromethylated *ortho*-nitrostyrenes. The second step is a one pot formation of α-CF_3_-β-(2-nitroaryl) enamines by the reaction with pyrrolidine. Finally, reduction of nitro group by Fe-AcOH-H_2_O system initiated intramolecular cyclization to form 2-CF_3_-indoles in up to 85% yields. A broad synthetic scope and simplicity of the procedures of all steps are the distinct advantages of the method. The prepared trifluoromethylated indoles are valuable staring materials to synthesize 3-functionalized derivatives using some reactions with C-electrophiles.

## Data Availability

The data presented in this study are available in Appendix A.

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
