# Peer review of "An Efficient Approach to 2-CF3-Indoles Based on ortho-Nitrobenzaldehydes"

_molecules, 2021, doi:10.3390/molecules26237365_

Round 1

Reviewer 1 Report

The manuscript presents an efficient approach to the synthesis of 2-CF3-indoles based on ortho 3-nitrobenzaldehyde. New compounds  are correctly characterized. In may opinion, the manuscript is well prepared and can be published in Molecules, after minor revision.

  1. Spectroscopic data of known compounds should be removed from the manuscript. They are presented in supplementary materials.
  2. Structural formulas of compounds 2a-2o and 4a-4m and a their data (Scheme 1 and 2 ) should be summarized in  the table (general formula and substituents, yield etc.).
  3. References - it requires harmonisation – lower and upper case were used for article titles.

Author Response

  1. Spectroscopic data of known compounds should be removed from the manuscript. They are presented in supplementary materials. done
  2. Structural formulas of compounds 2a-2o and 4a-4m and a their data (Scheme 1 and 2 ) should be summarized in the table (general formula and substituents, yield etc.). We believe, that current form of presentation is more clear for understanding
  3. References - it requires harmonisation – lower and upper case were used for article titles. done

Reviewer 2 Report

This manuscript submitted by Nenajdenko et al. mainly described a one-pot, two-step synthesis of 2-CF3-indoles. The synthetic sequence started with first catalytic olefination of 2-nitrobenzaldehydes with CF3CCl3 to yield stereoselectively trifluoromethylated o-nitrostyrenes. The subsequent treatment of these alkenes with pyrrolidine allowed the preparation of α-CF3-β-(2-nitroaryl) enamines. Final one-pot reduction of nitro-group by Fe-AcOH-H2O system initiated intramolecular cyclization afforded the target 2-CF3-indoles. This methodology provides alternative access to biologically important 2-CF3-indoles with relatively inexpensive reagents. If there is any drawbacks in this synthesis, I would say that 10 equivalents of pyrrolidine needed to be employed. Nevertheless, this disadvantage is well compensated by the utilization of widely commercially available o-nitroaldehydes as the starting material. Further, a number of different amine substituents can be incorporated onto the 5-position of 2-CF3-indoles by using 5-fluoro-2-nitrobenzaldehyde as the substrate, which might broaden their potential applications on biological activities as well as functional properties. Thus, the publication of this manuscript on the journal of Molecules is recommended. Some suggestions and comments for the authors are listed below.

  1. In Scheme 2, the bond between indole and CF3 of 4 is missing.
  2. In Scheme 4, Synthesis of enamine 3n and 2-CF3-3-benzylindoles 4n. I don’t see the 3-benzyl group on 4n.
  3. In Scheme 7, the solvent for the synthesis of compound 12 should be ethanol rather than methanol.

Author Response

  1. In Scheme 2, the bond between indole and CF3 of 4 is missing.

corrected

  1. In Scheme 4, Synthesis of enamine 3n and 2-CF3-3-benzylindoles 4n. I don’t see the 3-benzyl group on 4n.

corrected

  1. In Scheme 7, the solvent for the synthesis of compound 12 should be ethanol rather than methanol.

corrected

Reviewer 3 Report

The paper is well written and scientifically clear and relevant. Anyway some points that have to be taken care of:

  1. Trifluoromethylated styrenes were obtained as E/Z mixture and the ratios were determined by 19F NMR. How the authors determined that the major isomer was the Z? Please, add an explanation about the configuration assignment.
  2. Lines 84-86: from the text it seems that the yield of 62% is relative to the Wittig, but in Scheme 1 it is relative to the catalytic olefination reaction. Please, correct the text or the scheme.
  3. Please add a comment in the text about the mechanism proposed in scheme 3
  4. Line 157: in the text it is mentioned compound 2o, but the correct number is 2n
  5. Scheme 7. There are some errors. The formula of methanesulfonic acid is wrong, the correct one is MeSO3H and not Me3SO3 The solvent of the reactions to compounds 12 and 13 is EtOH and not MeOH
  6. Can the authors comment the mechanism leading to compound 13.
  7. Experimental part. In some cases, the eluent mixture of the chromatography is not reported when a general procedure is employed. Please add these data for each compound, when missing.

Author Response

1.Trifluoromethylated styrenes were obtained as E/Z mixture and the ratios were determined by 19F NMR. How the authors determined that the major isomer was the Z? Please, add an explanation about the configuration assignment.

added

2.Lines 84-86: from the text it seems that the yield of 62% is relative to the Wittig, but in Scheme 1 it is relative to the catalytic olefination reaction. Please, correct the text or the scheme.

corrected

3. Please add a comment in the text about the mechanism proposed in scheme 3

added

4. Line 157: in the text it is mentioned compound 2o, but the correct number is 2n

corrected

5. Scheme 7. There are some errors. The formula of methanesulfonic acid is wrong, the correct one is MeSO3H and not Me3SO3 The solvent of the reactions to compounds 12 and 13 is EtOH and not MeOH

corrected

6. Can the authors comment the mechanism leading to compound 13.

done

7. Experimental part. In some cases, the eluent mixture of the chromatography is not reported when a general procedure is employed. Please add these data for each compound, when missing.

added

Reviewer 4 Report

In this manuscript, the authors reported a library of indoles including bis-indole having CF3-group via trifluoromethlated styrene’s in one-pot reduction followed by intramolecular cyclization with the usage of ortho-nitrobenzaldehdes. They synthesized these trifluoromethylated styrenes by catalytic olefination reaction (COR) and Wittig reaction. Moreover, these heterocycles might be medicinally important and biologically worthy. Overall, the scholarly quality of this paper is not satisfactory for
acceptance in molecules in the current form. However, there are some additional remarks that need to be clarify before its publishing.

My comments, suggestions, and questions to authors:

  1. Even though the manuscript contains potentially interesting data but it is not well organized and contains a lot of severances. The numbering of
    formulas in the Schemes are chaotic. The plan of the accomplished results and their description is confusing and inappropriate.
  2. Why compound 1 appears just after the compound 2?  The numberings of
    the compounds in this manuscript are totally confusing (Figure 2 vs Scheme 1) as well as in Scheme 5 [4o vs 4p]
  3. Line 99 in page 4: the structure of compound 3a has not been shown
    anywhere and in last sentence 3a in 85, 86 and 85% yield???
  4. For ex; in Scheme 3, there is no numbering for the alkyne derivative?? from 2c and authors must explain detail mechanism in the text.
  5. For ex; in Scheme 3, 2n, 2o =?? And 3n, 4n?? 4o (Scheme 5) confusing to this referee.
  6. In Scheme R1 & R2 = ?? is it same amine fragments??
  7. In Scheme 2 & 6, compound 4 confusing and should be change accordingly.
  8. Author should include references (previous reports) for each protocol in figure 2.
  9. Line 128 in page 4: the structure of compound 5, 5a has not been shown
    anywhere.
  10. Author should cite reference for each protocol in figure 2.
  11. Line 148, 151, the font of “1 h” is seemingly different compared with the other and please check these typos overall MS file.
  12. In Scheme 1 conditions, time should be mention in both cases and other Schemes as well.
  13. In Scheme 1, remove space in % of yields, should be uniform throughout MS file.
  14. Authors placed general remarks (page 7, line 79) in both cases (Manuscript and SI file), it should be either MS or SI file not both cases and check it as per journal guidelines.
  15. In experimental, is it compound 2d obtained from 1c?? line 275-276/page 9 and encourage authors to please check these errors and recheck all NMR data with spectral throughout the manuscript.
  16. The authors reported bis-indole derivative 13, in medicinal Chemistry, there is a special need of NCE's and this should be mentioned in introduction (figure 1). The coverage of the relevant literature should be improved, for example on bis- indoles; doi:10.1002/cncr.23619, https://doi.org/10.1016/j.tetlet.2016.10.112;https://doi.org/10.3762/bjoc.10.228; Naturforsch. 2004, 59b, 1137–1142 should be cited.  

Author Response

  1. Even though the manuscript contains potentially interesting data but it is not well organized and contains a lot of severances. The numbering of
    formulas in the Schemes are chaotic. The plan of the accomplished results and their description is confusing and inappropriate.

corrected

2. Why compound 1 appears just after the compound 2?  The numberings of
the compounds in this manuscript are totally confusing (Figure 2 vs Scheme 1) as well as in Scheme 5 [4o vs 4p]

Numbering in Figure 2 was deleted

  1. Line 99 in page 4: the structure of compound 3a has not been shown
    anywhere and in last sentence 3a in 85, 86 and 85% yield???

The structure and yield of 3a,c,d

  1. For ex; in Scheme 3, there is no numbering for the alkyne derivative?? from 2c and authors must explain detail mechanism in the text.

Numbering was added. The mechanism was explained

  1. For ex; in Scheme 3, 2n, 2o =?? And 3n, 4n?? 4o (Scheme 5) confusing to this referee.

corrected

  1. In Scheme R1 & R2 = ?? is it same amine fragments??

corrected

  1. In Scheme 2 & 6, compound confusing and should be change accordingly.

corrected

  1. Author should include references (previous reports) for each protocol in figure 2.

done

  1. Line 128 in page 4: the structure of compound 55a has not been shown
    anywhere.

The structures of intermediates and by products as well as their 19 NMR data and yields (in the crude reaction mixtures) were added to SI file.

  1. Author should cite reference for each protocol in figure 2.

done

  1. Line 148, 151, the font of “1 h” is seemingly different compared with the other and please check these typos overall MS file.

corrected

  1. In Scheme 1 conditions, time should be mention in both cases and other Schemes as well.

done

  1. In Scheme 1, remove space in % of yields, should be uniform throughout MS file.

done

  1. Authors placed general remarks (page 7, line 79) in both cases (Manuscript and SI file), it should be either MS or SI file not both cases and check it as per journal guidelines.

General remarks were deleted from SI file

  1. In experimental, is it compound 2d obtained from 1c?? line 275-276/page 9 and encourage authors to please check these errors and recheck all NMR data with spectral throughout the manuscript.

corrected

  1. The authors reported bis-indole derivative 13, in medicinal Chemistry, there is a special need of NCE's and this should be mentioned in introduction (figure 1). The coverage of the relevant literature should be improved, for example on bis- indoles; doi:10.1002/cncr.23619, https://doi.org/10.1016/j.tetlet.2016.10.112;https://doi.org/10.3762/bjoc.10.228; Naturforsch. 2004, 59b, 1137–1142 should be cited

The coverage of the relevant literature was improved. The text about bis-indoles were added just after the scheme 7.

Round 2

Reviewer 4 Report

This manuscript has been adjusted by the authors according to suggestions, therefore, this reviewer recommend to its acceptance. But, there are some very minor changes required. for ex. Exp. conditions such as time should be uniform 3 h not 3h, please check it small typos entire draft and SI as well.